# Static and Vibration Analysis of Imperfect Thermoelastic Laminated Plates on a Winkler Foundation

**DOI:** 10.3390/ma18153514

**Published:** 2025-07-26

**Authors:** Jiahuan Liu, Yunying Zhou, Yipei Meng, Hong Mei, Zhijie Yue, Yan Liu

**Affiliations:** 1Department of Architectural Engineering, North China Institute of Aerospace Engineering, Langfang 065000, China; liujh@stumail.nciae.edu.cn (J.L.); abcdefg@stumail.nciae.edu.cn (H.M.); zj.yue.mce@outlook.com (Z.Y.); 2Hebei Key Laboratory of Trans-Media Aerial Underwater Vehicle, Langfang 065000, China; mengyp@stumail.nciae.edu.cn; 3Department of Aeronautics and Astronautics, North China Institute of Aerospace Engineering, Langfang 065000, China; 4Langfang Sunshine Construction Engineering Quality Testing Co., Ltd., Langfang 065099, China; 15311448733@163.com

**Keywords:** state-space method, Winkler foundation, imperfect interfacial bonding, thermo-mechanical coupling fields, static bending, modal analysis

## Abstract

This study introduces an analytical framework that integrates the state-space method with generalized thermoelasticity theory to obtain exact solutions for the static and dynamic behaviors of laminated plates featuring imperfect interfaces and resting on a Winkler foundation. The model comprehensively accounts for the foundation-structure interaction, interfacial imperfection, and the coupling between the thermal and mechanical fields. A parametric analysis explores the impact of the dimensionless foundation coefficient, interface flexibility coefficient, and thermal conductivity on the static and dynamic behaviors of the laminated plates. The results indicate that a lower foundation stiffness results in higher sensitivity of structural deformation with respect to the foundation parameter. Furthermore, an increase in interfacial flexibility significantly reduces the global stiffness and induces discontinuities in the distribution of stress and temperature. Additionally, thermal conductivity governs the continuity of interfacial heat flux, while thermo-mechanical coupling amplifies the variations in specific field variables. The findings offer valuable insights into the design and reliability evaluation of composite structures operating in thermally coupled environments.

## 1. Introduction

Plate structures are extensively utilized across diverse engineering disciplines, including civil and structural engineering, marine, aerospace, and mechanical engineering [1,2,3,4]. Driven by modern engineering demands, composite materials, particularly laminated composite plates, have seen widespread application due to their superior mechanical properties. However, these laminated plates are often susceptible to non-ideal behaviors such as interfacial debonding, shear slip, and abrupt changes in heat flux due to the mismatch in material properties at interlayer interfaces. These defects severely undermine structural stiffness and stability. Such issues frequently manifest in aerospace vehicles, multilayer circuit substrates, and sandwich composite panels, where interfacial degradation or thermal mismatch induces local stress concentration and dynamic instability.

To model plate-foundation interaction, numerous elastic foundation models have been developed, ranging from the classical Winkler model to more advanced Pasternak and viscoelastic formulations. The Winkler foundation represents the simplest approach for characterizing this interaction. Owing to its conceptual simplicity, computational efficiency, and reasonable accuracy, it has been widely adopted in engineering practice. Xue et al. [5] explored the post-buckling path and dynamic characteristics of the plates resting on a Winkler foundation in thermal environments, elucidating the thermal buckling mechanisms and their influence on dynamic behaviors. Kardooni et al. [6] conducted a thermo-mechanical analysis of a simply supported five-layer functionally graded (FG) sandwich plate resting on a Winkler foundation using the first-order shear deformation theory (FSDT). Celep et al. [7] examined dynamic and static responses of an elastic triangular plate on a unilateral Winkler foundation, focusing on forced vibrations under uniform and eccentric loads. Zhao et al. [8] applied the differential quadrature method to analyze the vibrations in porous two-dimensional FG microbeams on Winkler foundations, and investigated the influence of factors such as the dimensionless foundation modulus. Despite its prevalence, it often inadequately reflects real plate-foundation interaction. The Pasternak foundation model [9] is an extension of the Winkler foundation model that takes into account the shear interaction. Zakria et al. [10] investigated the vibration characteristics of a generalized thermoelastic microbeam supported by a Pasternak foundation using the dual phase lag (DPL) model. Daikh et al. [11] explored a new model of FG plate structures (FGPSs), resting on a modified four-parameter Winkler/Pasternak elastic foundation, and conducted a comprehensive parametric analysis of bending behavior sensitivity. By incorporating different physical effects (viscosity or nonlinearity), a more accurate description of the foundation response is achieved. Van Vinh et al. [12] developed a general viscoelastic Winkler-Pasternak foundation model to investigate the vibration behavior of functionally graded sandwich porous plates with different boundary conditions. Kumar et al. [13] studied the vibration performance of a rectangular FG plate with variable thickness, analyzing the influence of Winkle-Pasternak–Kerr foundation on natural frequencies.

Despite significant advancements in research, most studies still assume perfect interlayer bonding, implying that field variables at the interface are continuous. However, in practical engineering applications, due to manufacturing defects and wear accumulation over time, interlayer interfaces in laminated structures may sustain varying degrees of damage, a phenomenon known as the imperfect interface effect. This effect significantly impacts structural performance. Numerous studies have extensively investigated the impact of imperfect interfaces on structural behavior [14,15,16]. Liu and Lu [17] established a transfer matrix relationship linking the state vectors defined at inner and outer surfaces of laminated piezoelectric spherical shells using an imperfect interface model, thereby investigating its influence on free vibration characteristics. Zheng et al. [18] employed the Galerkin method to solve nonlinear equilibrium differential equations for MEE laminated beams with imperfect interface, analyzing how interface bonding strength affects the distributions of displacement, stress, electric potential, and magnetic potential along the beam thickness.

Many studies rely on simplified plate and shell theories, which, as noted by Chen and Lee [19], may lead to notable errors. Using a specialized transfer matrix, Wang and Pan [20] developed an exact solution for simply supported piezoelectric laminates with thermally induced interface defects. Chen et al. [21] employed a three-dimensional exact theory to study the bending behavior of multiferroic rectangular plates with interface imperfections. They introduced a generalized interfacial spring model to describe these imperfections and formulated a state-space approach integrating the interface transfer matrix. The state-space method is particularly suitable for solving laminated structures with complex interfacial and thermal boundary conditions, as it facilitates the derivation of exact or semi-analytical solutions. Compared to variational or numerical approaches, it offers superior capabilities in handling multilayer anisotropy, interfacial discontinuities, and thermal-mechanical coupling, especially when closed-form solutions are sought. Complementing these analytical methods, Khatir et al. [22] proposed a novel hybrid algorithm, Particle Swarm Optimization-YUKI (PSO-YUKI), which is based on enhanced optimization techniques and integrates experimental and numerical analyses to efficiently identify double cracks in carbon fiber-reinforced polymer (CFRP) cantilever beams. Overall, AI-related research provides valuable insights for the development of this field, offering potential strategies for areas such as digital twin technology and AI-assisted modeling for thermoelastic systems, data-driven prediction of composite structures under complex loads, and interface stiffness estimation.

Despite recent advances, most existing models treat interfacial imperfections either in the mechanical or thermal domain separately, with few offering analytical insight into their coupled effects on the behavior of laminated plates. This study focuses on laminated composite plates with imperfect interfacial bonding and thermal conductivity mismatch, resting on an elastic foundation. To overcome the limitations of conventional approaches, we develop a generalized thermoelastic model using the state-space method, accounting for foundation reaction forces, interfacial flexibility, and thermal conduction discontinuities. Closed-form analytical solutions are obtained for static bending and free vibration problems. This work explores the evolution patterns of coupled thermoelastic responses under varying system parameters, and establishes a comprehensive analytical framework for the accurate modeling and design of layered structural systems operating under complex service conditions.

## 2. Basic Equations

Consider a thermoelastic laminated plate, as illustrated in Figure 1. On the basis of the plane strain hypothesis, only the displacement components along the *x* and *z* axes are considered. The plate is supported by a Winkler foundation at its lower surface, while the upper surface remains free. The temperature distributions on the lower surface and upper surface are uniform, denoted as $T_{0}$ and $T_{1}$, respectively.

In the Cartesian coordinate system (x,y,z), the constitutive relation, heat conduction relation, equilibrium equation, and thermal energy equation for the plane strain laminated plate are as follows.$$\sigma_{x}=c_{11}\frac{\partial u}{\partial x}+c_{13}\frac{\partial w}{\partial z}-\beta_{1}T$$$$\sigma_{z}=c_{13}\frac{\partial u}{\partial x}+c_{33}\frac{\partial w}{\partial z}-\beta_{3}T$$$$\tau_{xz}=c_{55}(\frac{\partial u}{\partial z}+\frac{\partial w}{\partial x})$$$$P_{x}=-k_{11}\frac{\partial T}{\partial x},\;P_{z}=-k_{33}\frac{\partial T}{\partial z}$$$$\frac{\partial \sigma_{x}}{\partial x}+\frac{\partial \tau_{xz}}{\partial z}+F_{x}=0,\;\frac{\partial \tau_{xz}}{\partial x}+\frac{\partial \sigma_{z}}{\partial z}+F_{z}=0$$(1)$$\frac{\partial P_{x}}{\partial x}+\frac{\partial P_{z}}{\partial z}=0$$
where (u,w) and ($\sigma_{ij}$, $\tau_{ij}$), represent the displacement and stress components, respectively, T is the temperature difference relative to a referenced temperature in the stress-free state, $P_{i}\;(i=x,z)$ represents the heat flux, $c_{ij}$, $\beta_{ii}$, $k_{ii}$ are the elastic modulus, thermal elastic modulus, and heat conduction coefficients, respectively, and $F_{i}\;(i=x,z)$ the external body force.

If body forces are neglected, the state-space method can solve multilayer structural problems. The advantage of this method lies in the fact that its unified solution form remains invariant irrespective of the number of layers. Consequently, for structures comprising a greater number of layers, this method can reduce the dimensionality of the problem and improve computational efficiency. By choosing *z* as the primary unknown, the state equation in matrix form can be derived from Equation (1) [23] as follows(2)∂F(ζ)/∂ζ=U⋅F(ζ)
where U is the state matrix shown in Section A.1, and the expressions for the state vector $F={\left[u\;\sigma_{z}\;T\;\tau_{xz}\;w\;P_{z}\right]}^{T}$, as well as the two induced variables $\sigma_{x}$ and $P_{x}$, are provided in Section A.2.

## 3. Formula Derivation

For a laminated plate with simply supported boundaries, an exact solution can be derived.

The simply supported plate is governed by the following boundary constraints:(3)$$\sigma_{x}=w=0,\;(x=0\;\mathrm{or}\;x=l)$$

The temperature field conditions are as follows:(4)T=0, (x=0 or x=l)

To solve the boundary value problem described above, a dimensionless transformation is first applied. The state vector is assumed to be(5)$$\left\{\begin{array}{l} u\\ \sigma_{z}\\ T\\ \tau_{xz}\\ w\\ P_{z} \end{array}\right\}=\left\{\begin{array}{c} h\bar{u}(\zeta )\cos n\pi \xi \\ -c_{55}^{\left(1\right)}{\bar{\sigma }}_{z}(\zeta )\sin n\pi \xi \\ c_{55}^{\left(1\right)}/\beta_{3}^{\left(1\right)}\bar{T}\sin n\pi \xi \\ c_{55}^{\left(1\right)}{\bar{\tau }}_{xz}(\zeta )\cos n\pi \xi \\ h\bar{w}(\zeta )\sin n\pi \xi \\ \frac{c_{55}^{\left(1\right)}k_{33}^{\left(1\right)}}{\beta_{3}^{\left(1\right)}h}{\bar{P}}_{z}\sin n\pi \xi \end{array}\right\}e^{i\omega t}$$
where ξ=x/l and ζ=z/h are dimensionless coordinate variables, ω is the circular frequency, and the superscript (1) indicates that the parameter refers to the material properties of the first layer (i.e., the bottom layer, as shown in Figure 1). Quantities with a bar such as $\bar{u}(\zeta )$ represent the dimensionless state vector.

Substituting Equation (5) into Equation (2), the following dimensionless state-space formulation is obtained(6)$$\partial \bar{F}(\zeta )/\partial \zeta =U_{e}\cdot \bar{F}(\zeta )$$
where $\bar{F}={\left[\bar{u}\;{\bar{\sigma }}_{z}\;\bar{T}\;{\bar{\tau }}_{xz}\;\bar{w}\;{\bar{P}}_{z}\right]}^{T}$ represents the dimensionless state vector. The expression of the coefficient matrix $U_{e}$ is given in Section A.3. The dimensionless size is s=h/l, and dimensionless frequency is $\Omega^{2}=\rho^{\left(1\right)}\omega^{2}h^{2}/c_{55}^{\left(1\right)}$.

According to the matrix operation rules [24], the solution to Equation (6) is derived as(7)$$\bar{F}(\zeta )=\exp[U_{e}(\zeta -\zeta_{i-1})]\bar{F}(\zeta_{i-1})\;\left(\zeta_{i-1}\leq \zeta \leq \zeta_{i},i=1,2,\ldots N\right)$$
where $\zeta_{0}=0$, $\zeta_{i}=z_{i}/h=\Sigma_{j=1}^{i}h_{j}/h$, and $h_{j}$ is the thickness of the *j*-th layer.

Let $\zeta =\zeta_{k}$ in Equation (7), yields(8)$${\bar{F}}_{1}^{\left(k\right)}=\exp(Ah_{k}){\bar{F}}_{0}^{\left(k\right)}=M_{k}{\bar{F}}_{0}^{\left(k\right)}\left(k=1,2,\ldots ,N\right)$$

This equation represents the interlayer transfer relationship between the upper surface ${\bar{F}}_{1}^{\left(k\right)}$ and the lower surface ${\bar{F}}_{0}^{\left(k\right)}$ of the *k*-th layer.

If one replaces sinπξ by cosπξ, conversely, an alternative exact solution can be obtained in a similar manner from Equation (6). In this case, both ends are subjected to guided support boundary conditions.(9)$$\tau_{xz}=u=0\left(x=0\;\mathrm{or}\;x=l\right)$$

Therefore, the temperature *T* can be either a sine or a cosine function.

## 4. Imperfect Interface

At the interface between the *k*-th and *(k +* 1*)*-th layers, imperfect mechanical conditions are simulated by employing the spring layer model [21]$$\sigma_{z}^{\left(k+1\right)}=\sigma_{z}^{\left(k\right)}=[w^{\left(k+1\right)}-w^{\left(k\right)}]/R_{zz}^{\left(k\right)}$$(10)$$\tau_{xz}^{\left(k+1\right)}=\tau_{xz}^{\left(k\right)}=[u^{\left(k+1\right)}-u^{\left(k\right)}]/R_{xx}^{\left(k\right)}$$
where $R_{xx}$, $R_{zz}$ represent the flexibility coefficient for shear stress $\tau_{xz}$ and the normal stress $\sigma_{z}$, respectively.

We propose two types of imperfect interfaces for heat conduction [25,26]. For the low-thermal-conductivity condition,(11)$$P_{z}^{\left(k\right)}=P_{z}^{\left(k+1\right)},\;T^{\left(k\right)}-T^{\left(k+1\right)}=-R_{l}P_{z}^{\left(k\right)}$$

While for the high-thermal-conductivity condition,(12)$$T^{\left(k\right)}=T^{\left(k+1\right)},\;P_{z}^{\left(k+1\right)}-P_{z}^{\left(k\right)}=R_{h}\partial^{2}T^{\left(k\right)}/\partial x^{2}$$
where $R_{l}^{\left(k\right)}$, $R_{h}^{\left(k\right)}$ represent the flexibility coefficient for low- and high-thermal-conductivity conditions.

According to Equation (5), Equations (10) and (11), or Equation (12), the state vector at the interlayer interface can be organized into the following matrix form(13)$${\bar{F}}_{0}^{\left(k+1\right)}=P_{k}{\bar{F}}_{1}^{\left(k\right)}$$
where $P_{k}$ is the interfacial transfer matrix, representing the transfer relationship between the top surface of the *k*-th layer and the bottom surface of the *(k +* 1*)*-th layer. And the interface transfer matrix for the low-thermal-conductivity and high-thermal-conductivity condition, $P_{l}^{\left(k\right)}$, $P_{h}^{\left(k\right)}$ are defined as(14)$$P_{k}^{l}=\left[\begin{array}{cccccc} 1 & & & {\bar{R}}_{xx}^{\left(k\right)} & & \\ & 1 & & & & \\ & & 1 & & & {\bar{R}}_{l}^{\left(k\right)}\\ & & & 1 & & \\ & -{\bar{R}}_{zz}^{\left(k\right)} & & & 1 & \\ & & & & & 1 \end{array}\right]$$(15)$$P_{k}^{h}=\left[\begin{array}{cccccc} 1 & & & {\bar{R}}_{xx}^{\left(k\right)} & & \\ & 1 & & & & \\ & & 1 & & & \\ & & & 1 & & \\ & -{\bar{R}}_{zz}^{\left(k\right)} & & & 1 & \\ & & {\bar{R}}_{h}^{\left(k\right)} & & & 1 \end{array}\right]$$
where ${\bar{R}}_{xx}^{\left(k\right)}=c_{55}^{\left(1\right)}R_{xx}^{\left(k\right)}/h$, ${\bar{R}}_{zz}^{\left(k\right)}=c_{55}^{\left(1\right)}R_{xx}^{\left(k\right)}/h$, ${\bar{R}}_{l}^{\left(k\right)}=k_{33}^{\left(1\right)}R_{l}^{\left(k\right)}/h$, ${\bar{R}}_{h}^{\left(k\right)}=-R_{h}^{\left(k\right)}{\left(n\pi s\right)}^{2}/k_{33}^{\left(1\right)}h$.

Combining the interlayer transfer relationship (Equation (8)) and interface transfer relationship (Equation (13)), one obtains(16)$${\bar{F}}_{1}^{\left(N\right)}=H{\bar{F}}_{0}^{\left(1\right)}=\left(\prod_{j=N}^{1}M_{j}P_{j-1}\right){\bar{F}}_{0}^{\left(1\right)}$$
where H denotes the global transfer matrix for a laminate with an imperfect interface, and $P_{0}=I$. When laminated plates are perfectly bonded, all $P_{j}$ become units, yielding $H=\prod_{j=N}^{1}M_{j}$.

Considering the effect of the Winkler foundation [27], the boundary conditions at the top and bottom surfaces of the plate are as follows(17)$${\bar{\sigma }}_{z}(1)=0,\;{\bar{\sigma }}_{z}(0)=\bar{K}\bar{w}(0)$$(18)$${\bar{\tau }}_{xz}(1)={\bar{\tau }}_{xz}(0)$$
where $\bar{K}=Kh/c_{55}$ is the dimensionless foundation coefficient.

The temperature condition is $T=\bar{T_{0}}$ (z=0), $T=\bar{T_{1}}$ (z=h).

Here, it is assumed that all layers in the *n*-layer plate have equal thickness, with the bottom temperature variation denoted as ${\bar{T}}_{0}=c_{55}^{\left(1\right)}T_{0}/\beta_{3}^{\left(1\right)}$ and the upper surface temperature variation denoted as ${\bar{T}}_{1}=c_{55}^{\left(1\right)}T_{1}/\beta_{3}^{\left(1\right)}$; both ${\bar{T}}_{0}$ and ${\bar{T}}_{1}$ are uniform and dimensionless.

By substituting the different mechanical and thermal boundary conditions at *z = 0* or *h*, the exact solution for static bending can be obtained.(19)$${\left\{\begin{array}{c} \bar{u}\\ \bar{w}\\ {\bar{P}}_{z} \end{array}\right\}}_{\zeta =0}={\left[\begin{array}{ccc} H_{21} & H_{25}+H_{22}\bar{K} & H_{26}\\ H_{31} & H_{35}+H_{32}\bar{K} & H_{36}\\ H_{41} & H_{45}+H_{42}\bar{K} & H_{46} \end{array}\right]}^{-1}\left\{\begin{array}{c} -H_{23}\cdot {\bar{T}}_{0}\\ {\bar{T}}_{1}-H_{33}\cdot {\bar{T}}_{0}\\ -H_{43}\cdot {\bar{T}}_{0} \end{array}\right\}$$
where $H_{ij}$ is the element in the *i*-th row and *j*-th column of matrix H.

For the free vibration problem, the inhomogeneous terms on the right-hand side of Equation (19) are neglected. To ensure a non-zero solution for the unknowns, the determinant of the coefficient matrix must be zero, thereby obtaining the free vibration solution for the laminated plate without external load:(20)$$\left[\begin{array}{ccc} H_{21} & H_{25} & H_{26}\\ H_{31} & H_{35} & H_{36}\\ H_{41} & H_{45} & H_{46} \end{array}\right]{\left\{\begin{array}{c} \bar{u}\\ \bar{w}\\ {\bar{P}}_{z} \end{array}\right\}}_{\zeta =0}=0$$

If the effect of the Winkler foundation is considered, i.e., by incorporating the foundation coefficient to Equation (20), the vibration solution can be obtained.(21)$$\left[\begin{array}{ccc} H_{21} & H_{25}+H_{22}\bar{Κ} & H_{26}\\ H_{31} & H_{35}+H_{32}\bar{Κ} & H_{36}\\ H_{41} & H_{45}+H_{42}\bar{Κ} & H_{46} \end{array}\right]{\left\{\begin{array}{c} \bar{u}\\ \bar{w}\\ {\bar{P}}_{z} \end{array}\right\}}_{\zeta =0}=0$$

## 5. Numerical Results and Discussion

### 5.1. Validation

The computational cost of the proposed state-space method was assessed as a function of layer count *N*. The analysis considers a steel plate with the following material properties: $c_{11}=269.23\times 10^{9}$ N/m^2^, $c_{13}=115.38\times 10^{9}$ N/m^2^, $c_{33}=269.23\times 10^{9}$ N/m^2^, $c_{55}=76.92\times 10^{9}$ N/m^2^, ρ=7850 kg/m^3^, the thickness-to-width ratio s=h/l=0.1, n=1. Computer configuration: CPU: Intel Core i7-8750H, 3.6 GHz; RAM: 16 GB. As depicted in Figure 2, the computation time varies linearly with *N*, confirming favorable computational efficiency. The total runtime remains within constrained to practical limits (less than 25 s for *N* = 50), demonstrates that the method is suitable for layered structures and parametric studies. Crucially, the dimension of the state-space matrix does not change with an increase in the number of layers of the plate, ensuring computational costs increase only linearly without excessive escalation. Therefore, the method remains numerically stable within the computational domain.

To verify the accuracy of the program, a comparison with finite element results was conducted. This validation has been presented in our previously published article [28], The finite element model employed shell elements with a global mesh size of approximately 0.5 and 361 elements in total, ensuring reliable accuracy without excessive computational cost. No additional calibration was required, as the material and geometric parameters matched those of the analytical model. The maximum error between the finite element method (FEM) results and the analytical solutions is 9.81%. The errors arise from the fact that the finite element model can only represent a finite-length plate, rather than an infinite-length plate, thus leading to some discrepancies with the analytical solution for plane strain problems.

This example only compares the piezoelectric results with our elastic results, so material properties from Ref. [29] (listed in Table 1) are restricted to elastic parameters. Table 2 presents the computed lower-order frequencies for a two-layer BaTiO_3_/CoFe_2_O_4_ laminated plate (infinite in the *y*-direction) with s=0.1, considering both perfect ($\bar{R}=0$) and imperfect ($\bar{R}=0.1$) interface conditions. The results show a close agreement with Ref. [29], expect the last computed frequency. When $\bar{R}=0.1$, the natural frequency is lower than that of a perfect interface ($\bar{R}=0$) due to weakened interfacial stiffness. Consequently, the value marked with an asterisk (*) corresponding to $\bar{R}=0.1$ should be 4.8423924, which is less than 5.28208894 (the perfect interface frequency). Thus, the calculation error in Ref. [29] resulted in a relatively significant deviation for the frequency of 4.8423924. The comparison presented in Table 2 validates the correctness of the derivations and programming in this paper, thereby ensuring that the model can effectively analyze the static and dynamic characteristics of thermoelastic laminated plates, and investigate the influence of initial stresses and interface imperfections on various field variables.

### 5.2. Frequency Analysis

In this section, several numerical examples are presented to analyze the free vibration and static bending behavior of simply supported thermoelastic laminated plates. The solutions are obtained by solving the free vibration equations (Equations (20) and (21)) and static bending equation (Equation (19)) numerically to investigate parameter effects on plate response. The plates consist of two materials (Si_3_N_4_/Cobalt), the material properties [30] provided in Table 3. The dimensionless thickness-to-width ratio is s=h/l=0.1, and the dimensionless interlayer interface flexibility coefficients are ${\bar{R}}_{xx}^{\left(k\right)}={\bar{R}}_{zz}^{\left(k\right)}={\bar{R}}_{h}^{\left(k\right)}={\bar{R}}_{l}^{\left(k\right)}=\bar{R}$, and $\bar{T_{0}}=0$, ${\bar{T}}_{1}=1$.

Table 4 presents the results of the free vibration analysis of the laminated plates, illustrating the influence of five interfacial flexibility coefficients on the dimensionless natural frequencies. As shown, increasing these coefficients progressively reduces the natural frequencies. Higher interfacial flexibility corresponds to diminished structural stiffness, thereby lowering natural frequency. This reduction increases susceptiblity to large-amplitude vibrations and potential dynamic instability during operation.

Under the conditions in Table 4 with foundation coefficient $\bar{K}=200$, and the corresponding dimensionless vibration frequencies are listed in Table 5. The influence of the interfacial flexibility coefficients observed here is consistent with the trend reported in Table 4. Additionally, it is evident that increasing the foundation stiffness parameter induces a hardening effect, thereby enhancing the structural stiffness and consequently increasing the natural frequencies. Moreover, a comparison of the vibrational frequencies under low- and high-thermal-conductivity conditions reveals negligible differences, indicating minimal influence of thermal conductivity on the structural vibration characteristics within the dynamic model developed in this study. Unless otherwise specified, the subsequent analysis adopts a high-thermal-conductivity condition.

### 5.3. Modal Analysis

To assess the combined effects of imperfect interfaces and foundation stiffness on the vibration characteristics of thermoelastic laminated plates, the material parameters from Table 3 were employed to compute modal shapes of field variables. The results with and without foundation effects are shown in Figure 3 and Figure 4.

Figure 3 presents vibration characteristics of modal shapes in thermoelastic laminated plates with varying interfacial flexibility coefficients in the absence of a foundation. The corresponding dimensionless frequencies (with wave number *n* = 1) are 0.0379, 0.0377, 0.0373, and 0.0368, respectively. As shown, the field variables remain continuous under the perfect interface condition ($\bar{R}=0$), but as the interface flexibility coefficient increases, the continuity of the displacement fields across layers is progressively disrupted, exhibiting noticeable jumps at the interfaces in the modal shapes. Specifically, weakened interfacial stiffness reduces interlaminar shear force transmission, causing partial decoupling of adjacent layers. Discontinuities emerge as abrupt field variable changes at interfaces, with thermo-mechanical coupling amplifying this behavior through incompatible thermal expansions that intensify displacement gradients. Modal shape discontinuities correlate with localized stress concentration zones, accelerating fatigue crack initiation under thermal cycling. This mechanism accounts for the premature delamination observed in spacecraft multi-layer insulation systems subjected to orbital thermal cycling.

Under imperfect interface conditions, temperature and heat flux distributions exhibit interfacial phase reversal. This phenomenon arises due to phase shifts and coupling mismatches in thermo-mechanical vibration modes, which alter thermal responses and reverse local temperature gradient direction, thereby inducing thermal phase inversion. This phase reversal critically compromises spacecraft thermal protection systems by accelerating fatigue crack growth under thermal cycling.

Similarly, Figure 4 displays the modal shapes of the thermoelastic laminated plates supported by a Winkler foundation ($\bar{K}=200$), with corresponding dimensionless frequencies of 0.4402, 0.4389, 0.4342, and 0.4286 (wave number *n* = 1). Interfacial jumps in the displacement *u*-field intensify with increasing interfacial flexibility. However, foundation effects reduce the overall variation magnitude compared to Figure 3, attributable to vertical elastic support suppressing transverse displacements. Simultaneously, foundation-interface interactions amplify vertical displacement variations in modal shapes by enhancing localized responses near interfaces. Under thermo-mechanical coupling, other field variables exhibit minor but significant variations relevant to multi-physics engineering applications. Furthermore, the heat flux remains discontinuous across the interfaces, consistent with both the high-thermal-conductivity defect model (Equation (12)) and Figure 3 observations.

### 5.4. Static Bending Analysis

In order to investigate the effects of various parameters on the static bending behavior of the thermoelastic laminated plates, results are graphically presented. In the following cases, the heat flux discontinuity occurs at interfaces while temperature maintains continuity, consistent with the high thermal conductivity defect model (Equation (12)). Conversely, low-thermal-conductivity conditions yield opposite interfacial behavior (Equation (11)). This distinction will not be repeated in the subsequent discussions.

Figure 5 shows the static distribution of field variables through the thickness direction under high-thermal-conductivity conditions with different dimensionless foundation coefficients $\bar{K}$. Increasing $\bar{K}$ initially induces rapid reduction in stress and displacement fields. Beyond a critical $\bar{K}$ value, their influence progressively diminishes until values stabilize. This behavior indicates that the dimensionless foundation coefficient affects the stiffness and load-bearing capacity of the foundation, thereby directly influencing the force transfer between the structure and the foundation. Notably, the T and $P_{z}$ curves overlap, demonstrating negligible sensitivity to $\bar{K}$ variations. Since the impact of the dimensionless foundation coefficient on the field variables stabilizes after reaching a certain value, $\bar{K}=200$ is used in the following discussion unless otherwise specified.

Further analysis examines interfacial flexibility coefficient and upper-surface temperature effects on through-thickness field-variable distributions under different thermal conditions. A comparative analysis is also performed to highlight the differences in these distributions under high or low-thermal-conductivity conditions.

Figure 6 displays the distribution curves of field variables across the thickness for the thermoelastic laminated plate under low thermal conductivity with varying interface flexibility coefficients. Imperfect interface bonding significantly alters the distribution characteristics of the field variables: increasing $\bar{R}$ intensifies interfacial jumps in u and w, while simultaneously reducing $\sigma_{z}$ and w values at the upper surface. An increase in interfacial flexibility reduces the thermal stress at the top surface, which may help alleviate certain stress peaks; however, it also decreases the overall stiffness and load-bearing capacity, potentially leading to premature failure under high mechanical loads or thermal shocks, such as those encountered in engine compartments or satellite outer walls. Furthermore, due to interlayer shear slip, the shear stress distribution across thickness also exhibits variation.

Similarly, Figure 7 shows through-thickness field variable distributions under high-thermal-conductivity conditions. As interfacial flexibility coefficient $\bar{R}$ increases, the discontinuous jumps in displacements (u, w) and $P_{z}$ intensify at interfaces. In comparison with the case of low-thermal-conductivity (Figure 6), stress field variations substantially diminish under high-thermal-conductivity conditions, because under these conditions, efficient through-thickness heat transfer generates smoother temperature gradients and homogenizes the thermal field, thereby limiting stress distribution sensitivity to $\bar{R}$ variations. The homogenization of thermal conduction mitigates the influence of interfacial flexibility on structural stress distribution. This averaging effect enhances tolerance to manufacturing defects and aging-induced degradation, particularly in thermal barrier coatings subjected to cyclic thermal loading.

Furthermore, Figure 8 illustrates through-thickness field variables distributions in the thermoelastic laminated plate under varying upper surface temperatures. At perfect interface ($\bar{R}=0$), the field variables remain continuous at the interface. However, as the interface flexibility coefficient increases ($\bar{R}>0$), jumps in displacements (u, w) emerge at the interface, and they become more pronounced with increasing $\bar{R}$ (the subplot of w in Figure 7). Compared to Figure 7, elevated temperature ${\bar{T}}_{1}$ significantly amplifies field variable magnitudes at interface and upper surface. Temperature increases intensify absolute field variable values, enhancing distribution regularity due to thermal gradient amplification that strengthens thermoelastic coupling. The elevated thermal loads induce larger thermal expansions, and when combined with interfacial mismatches, lead to increased stress, displacement, and heat flux responses throughout the structure. This necessitates the incorporation of thermo-mechanical coupling simulations during the design stage to avoid underestimating risks associated with thermoelastic interactions.

Finally, Figure 9 presents the comparison of field variables across the thickness under high- or low-thermal-conductivity conditions, with $\bar{R}=0.1$. As shown, under otherwise identical conditions, higher thermal conductivity leads to greater variations in stress and displacement fields. This is primarily because enhanced heat conduction promotes the rapid and uniform transfer of thermal energy across the layers, thereby intensifying thermal gradients and thermal mismatch stresses. Consequently, the thermally induced deformations become more fully developed throughout the structure, especially in the presence of interfacial imperfections and heterogeneous material properties.

## 6. Conclusions

This paper systematically investigates the static bending and dynamic characteristics of laminated plates with imperfect interfaces resting on a Winkler foundation, utilizing a combined framework of the state-space method and generalized thermoelasticity theory. The main conclusions are as follows:(1)The presence of imperfect interfaces reduces the overall stiffness of the laminated plate, leading to a decrease in its natural frequency. However, the foundation effect provides additional elastic support, thereby increasing the overall vibration frequency of the structure. Under low thermomechanical coupling and uniform thermal fields, thermal conductivity has a limited influence on the vibrational response.(2)Under thermo-mechanical coupling, phase inversion phenomena are observed between temperature and heat flux in the modal patterns. Incompatible thermal expansions at weak interfaces intensify displacement gradients, leading to enhanced local deformation and stress concentration.(3)When the dimensionless foundation coefficient is small, its impact on deformation is more pronounced. As the dimensionless foundation coefficient increases, the changes in stress and displacement gradually diminish. When the dimensionless foundation coefficient reaches a specific value, the effect on field variables stabilizes.(4)In the static analysis, it is evident that, under otherwise identical conditions, higher thermal conductivity leads to more pronounced variations in stress and displacement. Furthermore, an increase in the top surface temperature systematically amplifies the variations in all field variables.

This study provides a high-precision closed-form solution suitable for multi-parameter control, elucidating the coupling relationship between interfacial effects and thermo-mechanical fields. Compared to the existing literature, it offers greater theoretical completeness and higher engineering applicability. The findings provide a novel theoretical basis for the design of laminated structures, interface control, and material selection under thermal environments. Moreover, the proposed framework can be readily extendable to scenarios involving transient or stochastic thermal-mechanical loads, which are common in aerospace and civil applications. This adaptability significantly enhances the model’s applicability to practical conditions where thermal cycling, impact loading, or variable boundary conditions compromise structural reliability. Future work will focus on extending the model to more realistic scenarios, such as incorporating transverse and in-plane temperature gradients, non-uniform thermal fields, and complex foundation conditions, including variable-stiffness or viscoelastic substrates.

## Figures and Tables

**Figure 1 materials-18-03514-f001:**
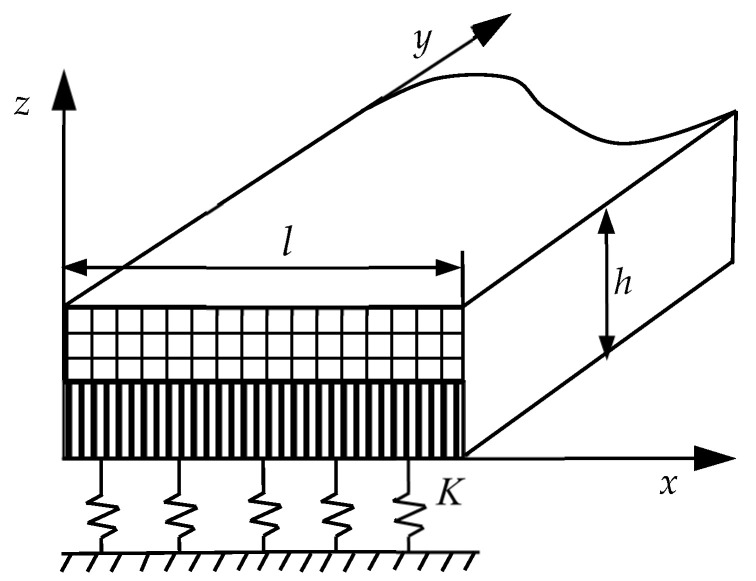
Thermoelastic laminated plates.

**Figure 2 materials-18-03514-f002:**
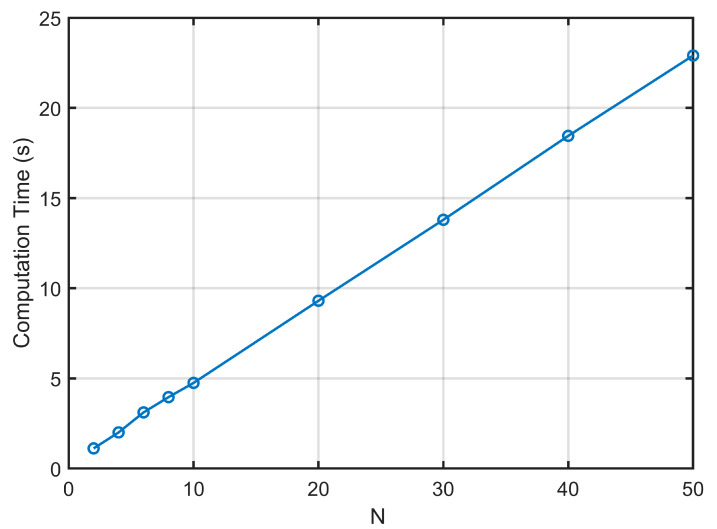
Computational cost.

**Figure 3 materials-18-03514-f003:**
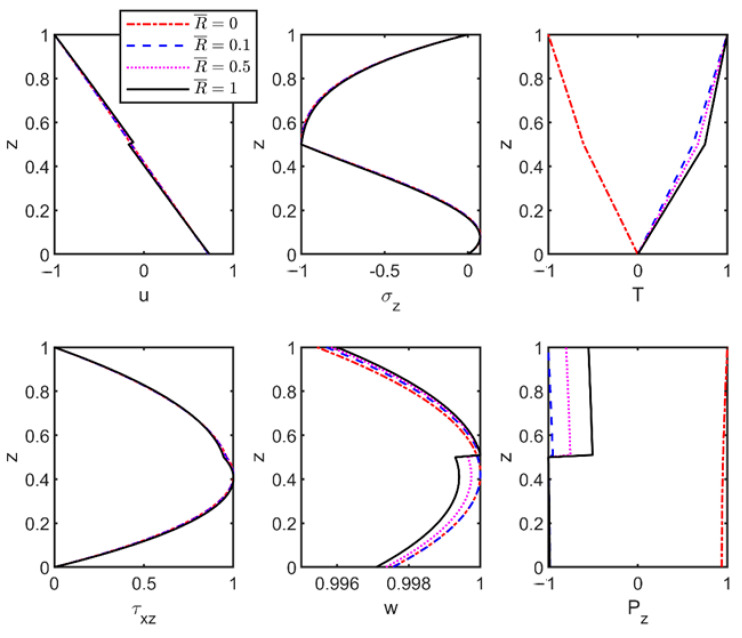
Effect of interface flexibility coefficient on the modal shapes of thermoelastic plates without Winkler foundation.

**Figure 4 materials-18-03514-f004:**
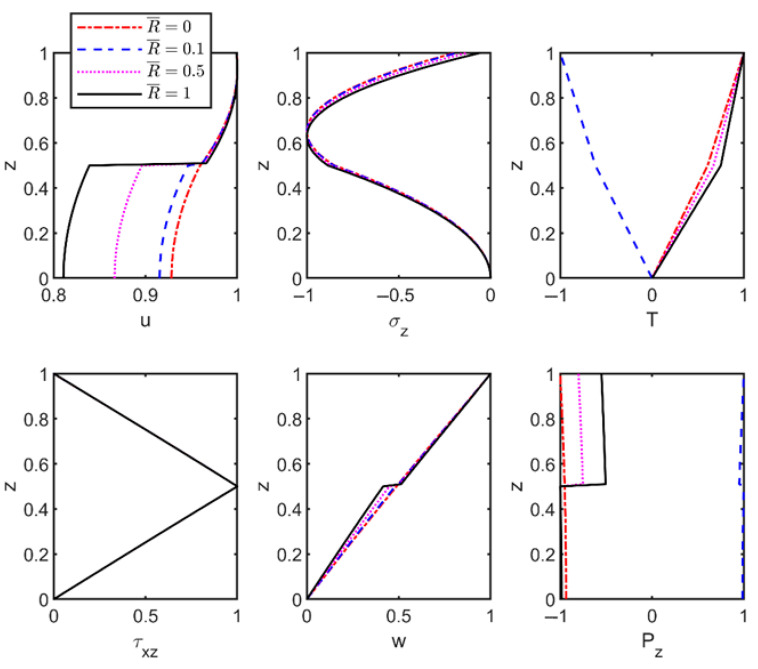
Effect of interface flexibility coefficient on the modal shapes of thermoelastic plates with Winkler foundation.

**Figure 5 materials-18-03514-f005:**
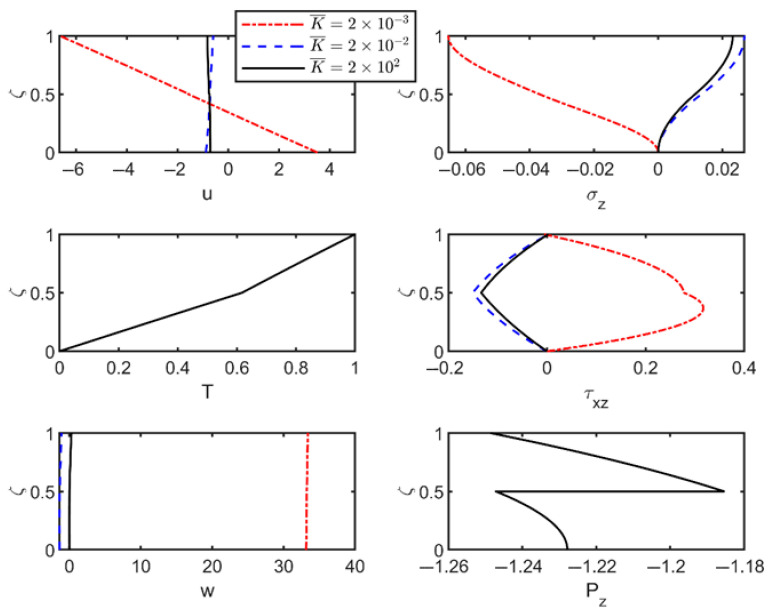
Effect of the variation in the dimensionless foundation coefficients on the through-thickness static distribution of field variables under high thermal conductivity.

**Figure 6 materials-18-03514-f006:**
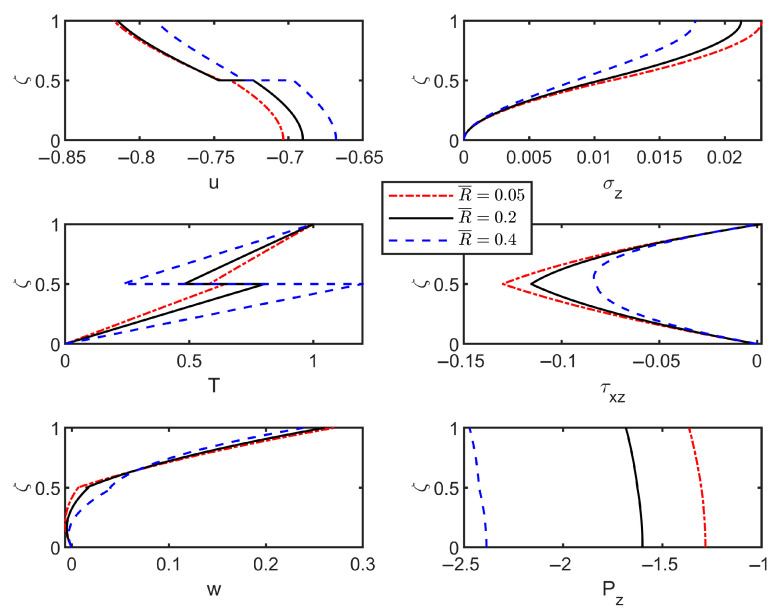
Effect of the variation in the dimensionless foundation coefficient on the through-thickness static distribution of field variables under low thermal conductivity.

**Figure 7 materials-18-03514-f007:**
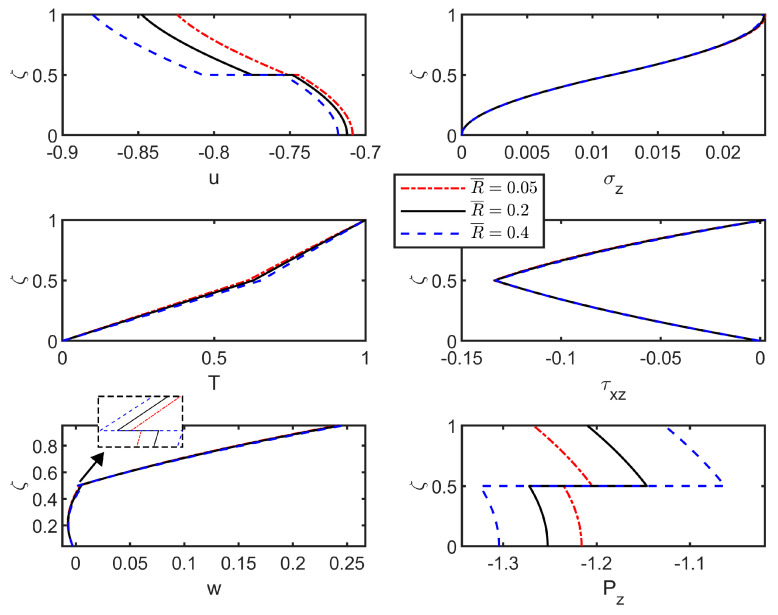
Effect of interface flexibility coefficient variation on the through-thickness static distribution of field variables under high thermal conductivity.

**Figure 8 materials-18-03514-f008:**
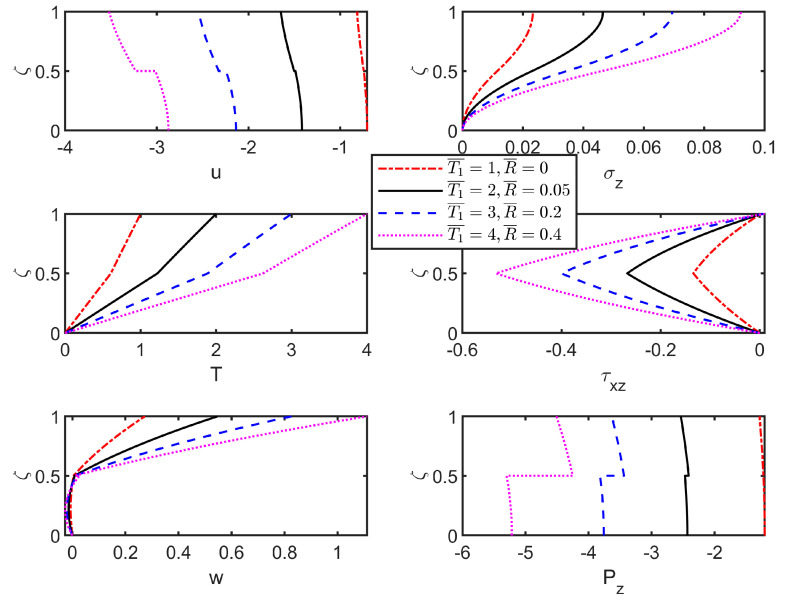
Effect of upper surface temperature variation on the through-thickness static distribution of field variables under high thermal conductivity.

**Figure 9 materials-18-03514-f009:**
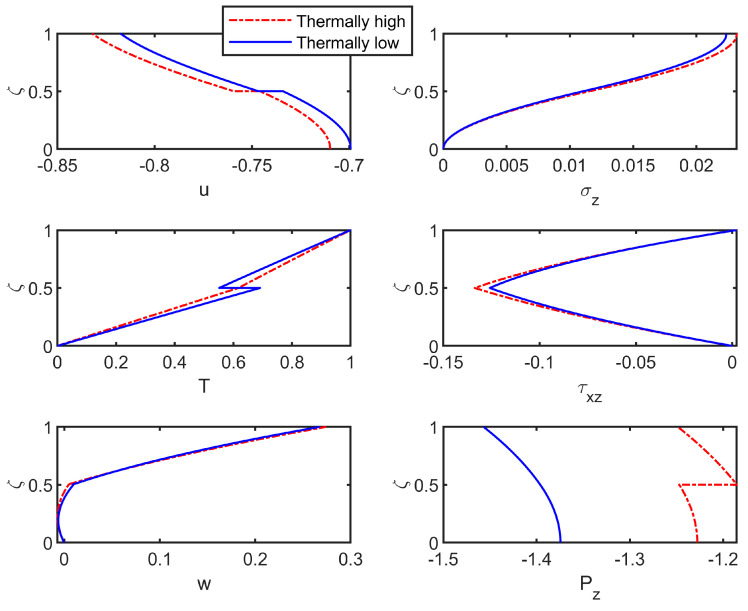
Field-variable distributions under high/low thermal conductivity.

**Table 1 materials-18-03514-t001:** Material properties (elastic properties only).

Material Properties		Material I (BaTiO_3_)	Material II (CoFe_2_O_4_)
Density (×10^3^ kg/m^3^)	ρ	5.85	5.3
Elastic constants (×10^9^ N/m^2^)	$c_{11}$	166	286
	$c_{13}$	77	173
	$c_{33}$	162	269.5
	$c_{55}$	43	45.3

**Table 2 materials-18-03514-t002:** A comparative analysis of the lower-order frequencies of a two-layer laminated BaTiO_3_/CoFe_2_O_4_ plate.

	$\bar{R}$ = 0			$\bar{R}$ = 0.1		
Mode	Ref. [29]	Present	Error (%)	Ref. [29]	Present	Error (%)
1	0.05331489	0.05338911	0.139	0.05314471	0.05322836	0.157
	0.60148603	0.60435096	0.476	0.60145326	0.60424158	0.464
	3.37140364	3.32553823	–1.360	3.04452649	3.02939519	0.497
2	0.20256454	0.20250937	–0.027	0.20032587	0.20040456	0.039
	1.19839311	1.2023193	0.328	1.19813853	1.20132375	0.266
	3.54649453	3.50637683	–1.131	3.23138724	3.22232972	–0.280
3	0.42370808	0.4227022	–0.237	0.41505409	0.41461008	–0.107
	1.78561828	1.78702302	0.079	1.78480695	1.78289915	–0.107
	3.80931899	3.77788484	–0.825	3.5098448	3.50995143	0.003
4	0.69261942	0.68949411	–0.451	0.67240478	0.67068738	0.255
	2.3569225	2.35060323	–0.268	2.35518253	2.33780214	–0.738
	4.13355602	4.11264194	–0.506	3.85138203	3.86009914	0.226
5	0.99129138	0.98482602	–0.652	0.955109	0.95130404	–0.398
	2.90417579	2.88382145	–0.701	2.90130344	2.84936174	–1.781
	4.49781778	4.48699559	–0.241	4.23358868	4.23334453	–0.006
6	1.30791029	1.29696028	–0.837	1.25267996	1.24595091	–0.537
	3.41658095	3.37589287	–1.191	3.41281569	3.29208185	–3.538
	4.88532846	4.87951409	–0.119	4.63907934	4.56049991	–1.694
7	1.63511459	1.61862439	–1.009	1.55908477	1.54854661	–0.676
	3.88061699	3.81512673	–1.688	3.87677183	3.6369564	–6.186
	5.28208894	5.26968072	–0.235	6.4357201 *	4.8423924	–24.76

**Table 3 materials-18-03514-t003:** Material properties.

Material Properties		Material I (Si_3_N_4_)	Material II (Cobalt)
Density (×10^3^ kg/m^3^)	ρ	3.2	8.836
Elastic constants (×10^9^ N/m^2^)	$c_{11}$	574	307.1
	$c_{13}$	127	102.7
	$c_{33}$	433	358.1
	$c_{55}$	108	75.5
Thermal moduli (×10^6^ N/K/m^2^)	$\beta_{1}$	3.22	7.04
	$\beta_{3}$	2.71	6.9
Heat conduction coefficients (W/K/m)	$k_{11}$	55.4	69
	$k_{33}$	43.5	69

**Table 4 materials-18-03514-t004:** The first five dimensionless natural frequencies of the laminated plates.

$\bar{R}$ = 0	$\bar{R}$ = 0.05	$\bar{R}$ = 0.1	$\bar{R}$ = 0.5	$\bar{R}$ = 1
0.0379	0.0378	0.0377	0.0373	0.0368
0.1420	0.1413	0.1406	0.1357	0.1357
0.2925	0.2899	0.2874	0.2706	0.2551
0.4402	0.4396	0.4389	0.4240	0.3948
0.4712	0.4652	0.4595	0.4342	0.4286

**Table 5 materials-18-03514-t005:** Effect of foundation on dimensionless natural frequencies in the laminated plates.

$\bar{R}$ = 0	$\bar{R}$ = 0.05	$\bar{R}$ = 0.1	$\bar{R}$ = 0.5	$\bar{R}$ = 1
0.4402	0.4395	0.4389	0.4342	0.4286
0.4402	0.8503	0.8461	0.8164	0.7747
0.4402	1.2168	1.2063	1.0246	0.7828
0.4402	1.5425	1.5223	1.0258	0.7886
0.4402	1.6933	1.5551	1.0313	0.7939

## Data Availability

The original contributions presented in this study are included in the article. Further inquiries can be directed to the corresponding author.

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
