# Peer review of "Static and Vibration Analysis of Imperfect Thermoelastic Laminated Plates on a Winkler Foundation"

_materials, 2025, doi:10.3390/ma18153514_

Round 1
Reviewer 1 Report
Comments and Suggestions for Authors
Dear Autors,
Please find comments on this article in the attachment.

Reviewer 2 Report
Comments and Suggestions for Authors
The present problem has many important aspects in scientific and practical recognition. I recommend publishing manuscript after some supplements:
1. The literature review should be enriched by the work of authors from the different countries.
2. The applications of the plate structure should be presented.
3. The introduction point should be presented with the descriptions: of the problem, the object of the analysis, the method of the solution, the aim of the investigations.
4. The explanation to the equation (4) should be presented.
5. The detailed description of the laminate material parameters should be presented, among them the configuration code.
6. The list of parameters should be added.
7. The reference to the accuracy analysis presented in [29] is not sufficient. Some more information and comparisons should be presented.
8. The future investigations should be presented.
Reviewer 3 Report
Comments and Suggestions for Authors
1- The study lacks experimental validation or benchmarking with real-world data. Though the model is said to be validated in a prior publication [29], it would strengthen this paper to summarize the key findings of that validation or replicate a simplified case.
2- The discussion would benefit from clearer physical interpretation of the vibration and bending responses. For instance, what does the variation in modal shapes imply for actual design or failure prediction in aerospace or civil components?
3- The introduction is comprehensive, but it would benefit from a clearer distinction between existing limitations in prior models and the specific novelties of your work.
4- The concluding section should be slightly expanded to emphasize the implications for real-life design of laminated plates especially under transient or stochastic loads.
5- This work could benefit from referencing recent methodologies that explore hybrid AI-based or optimization approaches in structural health monitoring and vibrational analysis of composite systems. These approaches can offer complementary tools for damage quantification, defect diagnosis, and structural response prediction under complex thermal-mechanical conditions, such us: [ A new hybrid PSO-YUKI for double cracks identification in CFRP cantilever beam, https://doi.org/10.1016/j.compstruct.2023.116803]. These studies provide a useful landscape of trends, including digital twin and AI-assisted modeling for thermoelastic systems, and highlight recent data-driven prediction for composite-enhanced systems under mechanical loads, and may suggest hybrid strategies for interface stiffness estimation.
6- Line 214: Clarify “validation has been presented in our previously published article [29]” either summarize findings briefly or mention key comparison metrics (e.g., RMS error or mode shape correlation).
7- Check for minor typos: e.g., "displacement equations for beam can be obtain" (line 44) → "can be obtained".
Round 2
Reviewer 2 Report
Comments and Suggestions for Authors
The manuscript has been supplemented with new content and corrections. Especially, accuracy comparison is very interesting. I recommend publishing in presented form.
Reviewer 3 Report
Comments and Suggestions for Authors
Authors have adressed required revisions and clarifications